# AC-ModNet: Molecular Reverse Design Network Based on Attribute Classification

**DOI:** 10.3390/ijms25136940

**Published:** 2024-06-25

**Authors:** Wei Wei, Jun Fang, Ning Yang, Qi Li, Lin Hu, Lanbo Zhao, Jie Han

**Affiliations:** School of Automation, Northwestern Polytechnical University, Xi’an 710072, China; wei_wei@mail.nwpu.edu.cn (W.W.); junfang@nwpu.edu.cn (J.F.); lqzy.77@gmail.com (Q.L.); linhu@mail.nwpu.edu.cn (L.H.); zhaolb724@mail.nwpu.edu.cn (L.Z.); han_jie@mail.nwpu.edu.cn (J.H.)

**Keywords:** molecular inverse design, AC-ModNet, specific intervals of attributes, adaptive balance factor, convolutional networks

## Abstract

Deep generative models are becoming a tool of choice for exploring the molecular space. One important application area of deep generative models is the reverse design of drug compounds for given attributes (solubility, ease of synthesis, etc.). Although there are many generative models, these models cannot generate specific intervals of attributes. This paper proposes a AC-ModNet model that effectively combines VAE with AC-GAN to generate molecular structures in specific attribute intervals. The AC-ModNet is trained and evaluated using the open 250K ZINC dataset. In comparison with related models, our method performs best in the FCD and Frag model evaluation indicators. Moreover, we prove the AC-ModNet created molecules have potential application value in drug design by comparing and analyzing them with medical records in the PubChem database. The results of this paper will provide a new method for machine learning drug reverse design.

## 1. Introduction

Discovering new molecules for drugs and materials can lead to enormous societal and technological advances [1]. However, comprehensive exploration of the vast number of potential chemical drugs is computationally difficult; estimates place the number of pharmacologically appreciable molecules at around $10^{23}$ to $10^{80}$ compounds [2,3]. Often, such searches are limited by the number of discovered structures and desired attributes such as solubility or toxicity. There are many approaches to explore chemical space in silico and in vitro, including high-throughput screening, combinatorial libraries, and evolutionary algorithms [4,5,6,7].

In the past few years, with the rapid development of deep learning and GPU (graphics processing units) [8] technologies, as well as the emergence of large molecular datasets (e.g., ZINC [9] and ChEMBL [10]), advances in machine learning (especially deep learning) methods have driven the design of new computing systems for modeling increasingly complex phenomena. Deep generative models constitute a method that has proven to be effective in modeling molecular data. Deep generative models have been widely used in a variety of fields, from generating synthetic images [11] and natural language text [12] to applications in biomedicine, including DNA sequence design [13], aging research [14], target identification [15], antimicrobial drug discovery [16], and drug repurposing [17,18]. An important application area of deep generative models is the reverse design of drug compounds for given attributes (solubility, ease of synthesis, etc.) [19].

There are many methods involved in the field of drug molecule reverse design in deep generative models, which can be divided into two categories. The first category is based on the VAE (Variational Autoencoder) [20], and the second category is based on GANs (Generative Adversarial Networks) [21]. The VAE (Variational Autoencoder) [20] is a structure in which variational reasoning is applied to generative models. Currently, a variety of VAE models have been applied to molecular feature extraction and have achieved certain results [22,23,24,25,26,27]. These works emphasize the accuracy of the molecular reconstruction and the ability to explore and inspire new molecules in potential continuous space, as well as optimize molecular specific attributes through Bayesian estimation. Kusner et al. [23] proposed the GVAE, which is an autoencoder based on context-independent molecular SMILES codes. Jin et al. [26] proposed the JT-VAE network, which achieves high-precision molecular reconstruction based on molecular graph encoding and molecular Christmas tree-assisted encoding. Jin et al. [27] later proposed the Hier-VAE model, which converts SMILES [28] into molecular graphs as input, learns substructure features through motif pre-extraction, simplifies the structure of the molecular generation model through three-layer decoding and encoding steps, and has excellent generation effects in the region of larger molecular weight. However, the VAE model lacks an accurate evaluation of the probability distribution, and the defects in its loss function calculation method often lead to unrealistic and ambiguous samples. In contrast, GANs (Generative Adversarial Networks) [21] have been proven to be an excellent generative model in the field of computer vision, and GANs have made some initial explorations in the field of molecular design [29,30,31,32]. Maziarka et al. [31] proposed a Cycle-GAN for molecular optimization tasks, which learns the characteristics of molecular transformation by corresponding to two types of disjoint molecules. Liu et al. [33] proposed a combination model of the VAE and GAN, MolFilterGAN, which aims to distinguish whether the molecules generated by the VAE network are biologically active drug molecules or molecules from the generated chemical space. Kadurin et al. [32] proposed an Adversarial Autoencoder (AAE) derived from the GAN architecture to identify and generate new compounds with predefined attributes. However, the input of this model is the fingerprint MACS [34], and the network consists only of a fully connected architecture, so it cannot effectively extract the complex features of the molecule.

Therefore, the above methods cannot perform interval classification-specific generation tasks for given attributes (solubility, ease of synthesis, etc.) [22,23,24,25,26,27,29,30,31,32,33]. Taking the solubility logP as an example, existing methods cannot generate molecules with the logP in a specific interval (0.0, 1.0]. To address this problem, the main research objectives of this paper are first to propose a novel AC-ModNet model, which effectively combines a VAE [20] with an AC-GAN [35] to generate molecular structures with specific interval functions; secondly, this model solves the problem of the weak generalization ability of VAE-like methods and the need for a large number of parameter adjustments. We can achieve the molecular effectiveness of the hierarchical VAE model [27] with 20 training parameter adjustments through two trainings; then we use convolutional neural networks to build the backbone of the GAN, which can effectively learn molecular graph structures and features; finally, we overcome the difficulty of GAN training by introducing new training loss functions and balance factors, we shorten the training time by an order of magnitude, and we solve the problem of mode collapse of the GAN method and the problem of weak model generalization ability. Our method uses three attributes—logP [36], SAScore [37], and QED [38]—which are classified into logp1–logp6, SA1–SA6, and QED1–QED7. By selecting the interval values, molecules that meet the corresponding conditions are generated.

We trained AC-ModNet on the open ZINC dataset [9], and the resulting generative model generated molecules with 100% validity, 98.64% average novelty, and 62.47% average uniqueness. We discuss the accuracy of the model through the comparison of the VAE [20], AAE [39], JT-VAE [26], hierarchical VAE [27], and LatentGAN [40] models using model evaluation criteria such as the FCD [41,42], IntDiv1 [43], IntDiv2 [43], SNN [44], Scaf [45], and Frag [46]. Our method performed best in both the FCD and Frag. In addition, by comparing and analyzing highly credible generated molecules with existing molecules on the PubChem dataset [47], we show that the molecules created by our model have the potential to be used in drug design. The results of this study will provide new avenues for machine learning drug reverse design and will also provide more reference methods for drug synthesis experiments.

## 2. Network

### 2.1. AC-ModNet

In the field of molecular design, we propose a method for effectively combining the VAE structure with the GAN structure. We chose the Hier-VAE [27] as the molecular encoder–decoder model used in this work.

As shown in Figure 1a, the model structure is divided into four parts: encoder $f_{encoder}$, decoder $f_{decoder}$, generator *G*, and discriminator *D*. The encoder extracts feature vectors from the training dataset. The generator generates feature vectors based on category labels. The discriminator judges and scores the samples conf∈[0,1], and the scores are passed back to the parameters for feedback learning. The decoder decodes the feature vector into a molecular graph *g* output. The data conversion in the encoder–decoder is shown in Equation (Equation 1).
(1)$$\begin{array}{c} \begin{array}{c} Conf_{real},C_{real}=D\left\{f_{encoder}\left(g_{real}\right)\right\}\\ Conf_{fake},C_{fake}=D\left\{G\left(c,\varepsilon \right)\right\}\\ g=f_{decoder}\left\{G\left(c,\varepsilon \right)\right\} \end{array} \end{array}$$

Figure 1b shows the model training process: the latent vector obtained from the training data (ground truth) through the encoder and the false sample vector obtained from the generator are both inputted into the discriminator to obtain the loss function. Figure 1c shows the generative model: the category tag $C=\{c_{1},c_{2},\cdots ,c_{n}\}$ which is added noise ε, is inputted into the generator, and the generated potential vector is decoded into a new molecule by the decoder.

In terms of the network structure design, considering the particularity of molecular graphs, we have introduced convolutional networks as category feature extraction networks in both the generator and discriminator. At the same time, the fully connected network is used as the mapping transformation between the molecular potential space and the quasi-image space.

### 2.2. Variational Autoencoders: VAEs

Due to the diversity and discreteness of molecular representation, Hier-VAE uses the Graph Neural Network (GNN) and Long Short-Term Memory (LSTM) [48] to extract the two-dimensional molecular representation feature *X* and map it to the potential vector *Z* using the VAE. Let *Z* obey the standard normal distribution p(Z)=N(0,I), and the mapping relationship is as follows:(2)$$p\left(X\right)=\sum_{z}p\left(X|Z\right)p\left(Z\right)$$

The Hier-VAE model is based on the structural motif for the hierarchical generation of the molecular diagram. Using the prelearning molecular motif, it can significantly improve the effectiveness of a molecular reconstruction and the accuracy of a molecular reconstruction with a large mass fraction. The hierarchical graph encoder maps the molecular graph to a continuous potential vector space through the Motif layer, Attachment layer, and Atomic layer three-layer coding structures (MPNs):(3)$$\begin{array}{c} h_{i}=f_{MPNs}\left\{e\left(a_{u}\right),e\left(b_{u\upsilon }\right),e\left(A_{i}\right),e\left(S_{i}\right)\right\}\\ Z_{g}=\mu \left(h_{1}\right)+exp\left(\sum h_{1}\right)\cdot \varepsilon \;\varepsilon ∼N\left(0,I\right) \end{array}$$
where $e\left(a_{u}\right),e\left(b_{u\upsilon }\right),e\left(A_{i}\right),e\left(S_{i}\right)$ define the embedding vector of the atoms in the molecular graph, $e(b_{u\upsilon })$ represents the embedding vector of the bonds in the molecular graph, $e(A_{i})$ represents the embedding vector of the junction in the molecular graph tree structure, and $e(S_{i})$ represents the embedding vector of the base order of the molecular connection tree. Through the information transmission and learning of the graph neural network, the characteristics of each base position in the tree structure are obtained. The potential vector is obtained through reparametric sampling using the features of the root order to represent the potential features of the molecular diagram.

The layered graphics decoder adopts a three-layer comprehensive prediction model of the Motif prediction, Attachment prediction, and graphics prediction. One motif and its connection position are predicted at a time (as shown in Equation (Equation 4)), and the tree-like structure of the motif tree is always maintained during the prediction and splicing process.
(4)$$p\left\{S_{t},\left(u_{i},v_{i}\right)\right\}=f_{VAE\_decode}\left(Z_{g},h_{t-1}\right)$$

By inputting the structural characteristics and molecular potential vectors of the connected molecules, the next motif type to be connected and the junction between the molecular part diagram and the motif are predicted in turn. It analyzes the final motif tree structure and outputs the ideal molecular diagram.

### 2.3. Generative Adversarial Networks

We use the Auxiliary Classifier GAN (or AC-GAN [35]) framework to generate the molecular potential vector of the specified attribute category, as shown in Figure 1. A continuous molecular attribute is divided into n intervals according to the value range and common values, and each interval corresponds to a category label *C*. The generator inputs a label and noise δ to generate sample $Z_{fake}=G\left(C,\delta \right)$, and it enters the potential vector corresponding to the real molecule and its corresponding category label into the discriminator for discrimination as P(T,C)|Z=D(Z). The objective functions of the network are as follows:(5)$$\begin{array}{c} max\;L_{D}=L_{T}+L_{C}\\ max\;L_{G}=L_{C}-L_{T}\\ L_{T}=E\left[logP\left(T=real|Z_{real}\right)\right]+E\left[logP\left(T=fake|Z_{fake}\right)\right]\\ L_{C}=E\left[logP\left(C=c|Z_{real}\right)\right]+E\left[logP\left(C=c|Z_{fake}\right)\right] \end{array}$$

The original generator loss includes the sample loss and the sample classification loss. The original discriminator loss consists of four parts: the truth loss, truth classification loss, false sample loss, and false sample classification loss. Considering the difficulty of training GANs, we have made in the following two improvements to the model.

#### 2.3.1. Loss Function Adjustment

In the training, the discriminator is prone to collude with the generator in classification tasks, thus resulting in a false sample classification loss returning to zero and preventing the generator from learning correct classification. It means that the sample classification loss of the generator is quickly set to zero. Therefore, we adjusted the discriminator’s training loss function (Equation (Equation 6)) to only learn the classification criteria of real data, thus ensuring that the discriminator learns the correct category features.
(6)$$L_{D}=E_{x∼P_{data}}\left[logD\left(x\right)\right]+E_{x∼G}\left[1-logD\left(x\right)\right]+E\left[logP\left(C=c|Z_{real}\right)\right]$$

In the later stage of training, the model was focused on exploring the edge space. In order to evenly distribute the generated molecules in the potential vector space, we added the Euclidean distance function of the truth samples. Its corresponding weight λ was set as a small value, which encourages the generator to prioritize learning in the space near the ground truth, thus achieving the goal of exploring the potential space of the molecules reasonably. The improved generator loss function is shown in Equation (Equation 7).
(7)$$L_{G}=E_{x∼G}\left[logD\left(x\right)\right]+E\left[logP\left(C=c|Z_{fake}\right)\right]+\lambda {∥x_{data}-x_{G}∥}_{2}^{2}$$

We verify the correctness of the adjusted loss function in Section 3.3.

#### 2.3.2. Adaptive Balance Factor

Generally, the learning rates of the discriminator and generator are fixed. However, the learning efficiencies of these two different networks vary during the whole process. If the learning rate does not match the current learning efficiency, it leads to inbalance in the discriminator and generator, and such an inbalance becomes even worse with ensuing iterations, which ultimately results in training failure. Hence, we introduced an adaptive “balance facto” α, which adaptively adjusts the learning of the generator and discriminator according to the current learning performance that can be measured by the loss function.

We designed a function the that evaluates the degree of inbalance in the discriminator and generator, which takes the loss functions of these two networks and outputs the balance factor α in [0, 1]. Then, the iteration steps of the discriminator or generator were adjusted based on the factor to make the learning efficiencies of them well-matched. The whole process is described as follows (Algorithm 1):    
**Algorithm 1:** Adaptive Training
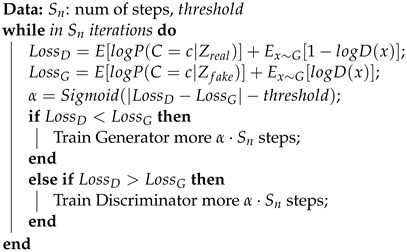


We validate the loss convergence rate after adding the balance factor in Section 3.4.

## 3. Experiment

In the implementation of the AC-ModNet model illustrated in Figure 1, we chose the Hier-VAE [27] as the molecular encoder–decoder model. The generator and discriminator of the AC-ModNet were both based on convolutional neural networks.

Generator network details: We embedded the category information as a vector of length 50, and we input the vector into the generator with noise. The input conversion interface of the generator consists of a two-layer FCN (fully connected network) with the Tanh activation function, where the length of the hidden layer is 128. It maps the molecular feature vectors into a 64∗64∗3 size image matrix, which facilitates subsequent convolutional learning. The convolutional unit of the generator consists of a convolutional layer, a LeakyReLU activation function, and a BatchNorm layer. The generated convolutional network consists of four convolutional units, with the first two each having an downsample layer finally outputting a 16∗16∗30 feature matrix. The feature matrix was straightened and mapped into a generated vector with a length of 250 through a two-layer FCN, where the middle layer has a length of 500.

Discriminator network details: The molecular feature vector with a length of 250 obtained through the encoder or the generator was inputted into the discriminator. We used a two-layer FCN to map the molecular feature vectors into a 64∗64∗3 size image matrix, where the hidden layer vector has a length of 512. Compared to the generator, the discriminator’s convolutional unit incorporates an additional dropout layer (dropout ratio of 0.25). The discriminative convolutional network consists of 4 convolutional units and a residual network, thus outputting an 8∗8 matrix with a depth of 128. The depth image was straightened into a vector, and the output part was composed of two different FCNs. One two-layer FCN maps the vector to a confidence in [0, 1] through the sigmoid activation function. Another two-layers FCN maps the vector to output discriminant values with class quantity length using the softmax activation function.

We used Python 3.8 to build the AC-ModNet, speed up the operation based on the 11.6 CUDA module, and perform molecular fingerprint calculation and molecular graph analysis by using the 2020.09.1.0 Rdkit [49] chemical toolkit.

### 3.1. Data Processing

The AC-ModNet is evalauted using the ZINC-250K open source dataset [9]. We used the molecular SMILES code and its three common important attributes—the logP, QED, and SAscore—which were input into the network after data cleansing and data balancing.

In order to obtain a rich set of recognizable motifs, we used the encoder–decoder model, which was pretrained in the ChEMBL dataset [10] with approximately 1800 K molecular units. Meanwhile, we removed 1936 molecular units, which could not be compiled by the encoder in the ZINC dataset in the data cleaning process. These molecules only account for 0.78% of the total data.

logP [36]: The lipid–water partition coefficient was proposed by Scott and Gordon to measure the lipophilicity (degree of fat solubility) of molecular compounds. Generally, when oral drugs penetrate through passive diffusion, a logP at 0–3 is considered as the best absorption range of the human body. High logP compounds have poor water solubility, and low logP compounds have poor lipid permeability. Therefore, we classified the low-value interval (−∞,0) into one category and the high-value interval (4,∞) into one category. The range of the logP with high bioavailability was more finely divided into six types of lipid–water partition coefficients.

Quantitative estimate of drug likeness (QED) [37]: Tian, Sheng, et al. proposed to use the important attributes of molecules to evaluate the drug-like attributes of molecules. This property ranges at (0, 1): a higher value indicates that more molecules evaluated by combining multimolecular descriptors fit the drug specification. Based on the need for high drug similarity and the distribution of this property under the Zinc 250 K dataset, we categorized low drug similarity at (0, 0.5) into one category, and the high drug similarity partition interval was set at 0.1. It was divided into six categories.

SAscore (SAS) [38]: This method, proposed by Ertl and Schuffenhauer [38], is composed of the fragmentScore and complexityPenalty. This method characterizes molecular synthesis accessibility as a fraction between 1 (easy to synthesize) and 10 (difficult to synthesize). Although the SAscore is widely distributed in the range of [2, 8], the distribution of the synthesis accessibility in biomolecules and recorded molecules is mainly concentrated in the range of [2, 4.5]. Therefore, we classified the extremely easy synthesis interval [1, 2] into one category, the difficult synthesis interval [4.5, 10] into one category, and we set the partition interval of the easier synthesis drug interval to 0.5. This attribute was divided into seven categories, and the data were balanced using the oversampling method. See Table 1 for more details.

### 3.2. Analysis of Backbone Network

The network structure has a significant impact on the molecular generation. The parameter effects obtained from the same throughput data of different networks also vary greatly.

We constructed two types of kernels—a fully connected network (model 1) and a convolutional network (model 2)—and we compared the convergence results of the model after training with the same data for equal iterations. Model 2 is the network described at the beginning of this section, and model 1 is the network that replaces the convolutional network and the first FCN connected to the start and end of the convolutional network with a 256∗256∗128 FCN, which also has a similar residual and dropout structure. Given a specific category, its accuracy is the ratio of the number of generated molecules in the the category to the total required amount. We generated 10k molecules for each category, and the accuracy of each category of the two molecular generated models is described in Table 2.

Observing the table, the accurate proportion of the molecules generated by the FCN is very low, which is close to the distribution of randomly generated molecules in the corresponding attribute categories. On the contrary, the accuracy of the generative model constructed by the CNN is significantly higher. The reason is that molecules have similar characteristics of graphs, and they can be well analyzed and learned using a convolutional neural network. Although the fully connected network is more universal compared to CNN, it is more difficult to be trained under the same situation.

Furthermore, we can observe that the relative accurary ratio of the generated molecular units across all the categories of model 2 tilts toward the distribution of the training data, as shown in Table 1. This makes sense, since the model can learn more for a category by providing more data belonging to it, which proves the learning effectiveness of our model with the backbone of the convolutional neural network.

### 3.3. Evaluation of Loss Function Adjustment

In the experiment, we firstly tried to use the original loss function for training, and we observed that the discriminator lost its rate of convergence quickly, but the generator lost no sign of convergence, which ultimately led to training failure. Hence, we have made some improvements to the generator and discriminator objective functions, which are described in Section 2.3.1.

Based on the adjusted loss functions, we trained each of the three molecular attributes 10 times and calculated the average loss curve of the network discriminator, as shown in Figure 2. It can be seen that the losses in the model training process could be effectively converged, thus indicating that the discriminator could effectively extract the associated features between the molecular structure and the attributes using the adjusted loss functions.

### 3.4. Evaluation of Adaptive Balance Factor

We introduced an adaptive “balance factor” that automatically adjusts the number of iterations for the discriminator and generator in a single loop based on loss, as described in Section 2.3.2. Figure 3 shows the accuracy change curve of the generator generating molecules for the specified category without and with the balance factor. It can be seen that, compared to the model collapse process where the accuracy dropped sharply after severe oscillation, the accuracy of the generator generating molecules with the addition of balance factors could be steadily improved.

The objective function of the whole network consists of the sum of two crossentropy losses in the range [0,∞], so we set the loop stop condition to the loss value 0.5, which is relatively small enough for the objective function to stop convergence. The training epoch of the AC-ModNet only needed one to two epochs to achieve the desired effect, which is one order of magnitude less than the epochs required for VAEs (as shown in Table 3). This shows that the “balance factor” significantly improved the training speed of the generative model.

## 4. Results and Discussion

### 4.1. Model Evaluation

We will evaluate our model from two aspects. The first aspect is to use three indicators (validity, novelty, and uniqueness) [26] to evaluate generated molecules in the required categories. The second aspect is to compare related models such as the VAE [20], AAE [39], JT-VAE [26], Hier-VAE [27], and LatentGAN [40] models using model evaluation criteria such as the FCD [41,42], IntDiv1 [43], IntDiv2 [43], SNN [44], Scaf [45], and Frag [46].

#### 4.1.1. Validity, Novelty, Uniqueness

We specified the logP, QED and SAScore attribute categories and generated 10K molecules for each required category; three indicators (validity, novelty, and uniqueness) [26] were used to evaluate the generated molecules in the required categories. Validity: A generated molecular is valid if its structure conforms to the basic rules of the molecular diagram, which are determined by the RDkit tool. Novelty: This defines the ratio of the number of generated molecules that are not in the training dataset to the number of whole generated molecular units. It indicates the greed and effectiveness of the network to explore the potential space of the molecules. Uniqueness: This defines the ratio of the number of nonrepeatable molecules generated by a network to the total number of generated molecules.

Table 4 shows the averaged validity, novelty, and uniqueness of the generated data for each attribute category. It can be observed that the generated molecules are all 100% effective molecules, thus indicating that the multilevel molecular graph decoder is very robust due to splicing with the GAN network that we designed, and it demonstrates its molecular reconstruction ability.

A high novelty value, which was greater than 98%, indicates that the generator excels in multidimensional exploration. This helps to avoid excessive dependence on certain structural types and facilitates the discovery of new structure–attribute relationships. In addition, a high novelty generative model helps to promote novelty and innovative research directions, which may bring new drug candidates, functional materials, or applications in other fields.

The average uniqueness of the generated molecules was about 63%, which means there are some repeatable molecules in the overall generated samples. the reason for this phenomenon is that we focused on the overall training process of the network. After increasing the number of training sessions, the discriminator loss converged, and when the network emphasizes improving classification accuracy, it will repeatedly generate molecules with high accuracy. Hence, it is easy to generate molecules with similar potential space when molecules are generated frequently, thus decoding them into same molecules.

We further analyzed the uniqueness with different generated numbers of molecules. The experimental results in Table 5 show that the uniqueness of the AC-Modnet was relatively stable for different scales of generation size. Thus, in practice, we can generate the required number of nonrepeatable molecules by increasing the total number of generated molecules.

In order to better observe the generated molecules, we sorted them according to their confidence value produced by the discriminator, and we took the top 10 of each category as examples for visualization, as shown in the Appendix B.

#### 4.1.2. Performance Metrics for Models

We compared the related models such as the VAE [20], AAE [39], JT-VAE [26], Hier-VAE [27], and LatentGAN [40] models using model evaluation criteria such as the FCD [41,42], IntDiv1 [43], IntDiv2 [43], SNN [44], Scaf [45], and Frag [46], which can be seen in Table 6.

**The Fréchet ChemNet Distance (FCD)** [41,42] is calculated using the activations of the penultimate layer of a deep neural network ChemNet trained to predict the biological activities of drugs. We computed the activations for canonical SMILES representations of the molecules. These activations capture both the chemical and biological properties of the compounds. For two sets of molecules *G* and *R*, the FCD is defined as
$$\mathrm{FCD}\left(G,R\right)={∥\mu_{G}-\mu_{R}∥}^{2}+\mathrm{Tr}\left(\Sigma_{G}+\Sigma_{R}-2{\left(\Sigma_{G}\Sigma_{R}\right)}^{1/2}\right)$$
where $\mu_{G}$ and $\mu_{R}$ are mean vectors, and $\Sigma_{G}$ and $\Sigma_{R}$ are full covariance matrices of the activations for molecules from sets *G* and *R*, respectively. The FCD correlates with other metrics. The values of this metric are non-negative, and a lower value is better.

**The Internal Diversity (IntDivp)** [43] assesses the chemical diversity within the generated set of molecules *G*.
$${\mathrm{IntDiv}}_{p}\left(G\right)=1-\sqrt[{p}]{\frac{1}{{\left|G\right|}^{2}}\sum_{m_{1},m_{2}\in G}T{\left(m_{1},m_{2}\right)}^{p}}$$
This metric detects a common failure case of generative models—mode collapse. With mode collapse, the model produces a limited variety of samples, thus ignoring some areas of the chemical space. A higher value of this metric corresponds to higher diversity in the generated set. In the experiment results, we report the ${IntDiv}_{1}\left(G\right)$ and ${IntDiv}_{2}\left(G\right)$. The limits of this metric are [0; 1].

**The Similarity to a Nearest Neighbor (SNN)** [44] is an average Tanimoto similarity $T\left(m_{G},m_{R}\right)$ (also known as the Jaccard index) between the fingerprints of a molecule $m_{G}$ from the generated set *G* and its nearest neighbor molecule $m_{R}$ in the reference dataset *R*:$$\mathrm{SNN}\left(G,R\right)=\frac{1}{\left|G\right|}\sum_{m_{G}\in G}\max_{m_{R}\in R}T\left(m_{G},m_{R}\right)$$
In this work, we used standard Morgan (extended connectivity) fingerprints with a radius of two and 1024 bits computed using the RDKit library. The resulting similarity metric can be interpreted as precision: if the generated molecules are far from the manifold of the reference set, the similarity to the nearest neighbor will be low. The limits of this metric are [0; 1].

**The Fragment Similarity (Frag)** [46] compares the distributions of BRICS fragments in the generated and reference sets. We denote $c_{f}\left(A\right)$ as the number of times a substructure *f* appears in the molecules from set *A*, and we denote the set of fragments that appear in either *G* or *R* as *F*; the metric is defined as a cosine similarity:$$\mathrm{Frag}\left(G,R\right)=\frac{\sum_{f\in F}\left(c_{f}\left(G\right)\cdot c_{f}\left(R\right)\right)}{\sqrt{\sum_{f\in F}c_{f}^{2}\left(G\right)}\sqrt{\sum_{f\in F}c_{f}^{2}\left(R\right)}}$$ If the molecules in both sets have similar fragments, the Frag metric is large. If some fragments are over- or under-represented (or never appear) in the generated set, the metric will be lower. The limits of this metric are [0; 1].

**The Scaffold Similarity (Scaff)** [45] is similar to the Fragment Similarity metric, but, instead of fragments, we compare the frequencies of Bemis–Murcko scaffolds. A Bemis–Murcko scaffold contains all the molecule’s ring structures and linker fragments that connect the rings. We used the RDKit implementation of this algorithm, which additionally considers carbonyl groups attached to rings as part of a scaffold. We denote $c_{s}\left(A\right)$ as the number of times a scaffold *s* appears in the molecules from set *A*, and we define the set of fragments that appear in either *G* or *R* as *S*; the metric is defined as a cosine similarity:$$\mathrm{Scaf}\left(G,R\right)=\frac{\sum_{s\in S}\left(c_{s}\left(G\right)\cdot c_{s}\left(R\right)\right)}{\sqrt{\sum_{s\in S}c_{s}^{2}\left(G\right)}\sqrt{\sum_{s\in S}c_{s}^{2}\left(R\right)}}$$
The purpose of this metric is to show how similar the scaffolds present in generated and reference datasets are. For example, if the model rarely produces a certain chemotype from a reference set, the metric will be low. The limits of this metric are [0; 1]. Note that both the fragment and scaffold similarities compare molecules at a substructure level. Hence, it is possible to have a similarity of one, even when the *G* and *R* contain different molecules.

Through such comparative experiments, we found that our model performed best in the FCD, thus indicating that our model has better effectiveness, as well as chemical and biological significance, than common methods. In terms of the Frag score, the results of our method and mainstream methods such as the VAE are very close, thus indicating that the Fragment Similarity of the molecules generated by our method and mainstream methods is relatively good.

### 4.2. Potential Application Value of Generated Molecules

For the purpose of evaluating the application value of our generated molecules, we compared them with recorded ones in the PubChem [47], which is a database of organic small molecule biological activity data. The database records existing and tested molecules; it currently contains 112M compounds, 297M substances, and 301M biological activities.

For each category, we ranked the molecules according to the confidence level, which is the molecular discriminant weight. Then, we took the top-2 molecules of each category to search for similar ones in the PubChem pharmaceutical database for comparison. Finally, we calculated the similarity between the new molecules and the applied molecules in the retrieval using the MACCS fingerprint and the Dice similarity method. Selected results of these high confidence generated molecules and the similar ones in the PubChem are listed in the Appendix A.

As can be seen, a similarity = 1.00 indicates that the generated molecules exist in the database. A considerable portion of the generated molecules in the table can be searched in this database to obtain one-to-one corresponding results. Furthermore, other new molecules can also find highly similar organic molecules, thus indicating that they are potential optimized molecules or byproducts with biased attributes.

Meanwhile, for the observation of approximate molecules, the molecular structure and the interactions between its functional groups influences the attributes of the molecule, so multiple optimization schemes for approximate molecules can be obtained. In the following, we will analyze some examples of each property; as well, the values of the logP, QED, and SAScore were calculated using Rdkit [49].

In order to evaluate the application value of the molecules we generated, we compared them with the molecules recorded in PubChem, which is a database of organic small molecule bioactivity data. This database records existing and tested molecules, and it currently contains 112M compounds, 297M substances, and 301M biological activities. Figure 4 compares the molecules generated by our model with real drug molecules in the PubChem dataset. Our method can produce drug molecules that actually exist in reality, which indirectly shows that our method is effective. In addition, we put several tables in the Section A.1. These tables are part of our generated results. We summarize them according to similarity and confidence. We hope that the results of this part can contribute to drug synthesis work.

For example, in Section A.1, which list examples of the generated molecules in logP attributes, the generated molecule CC(=O)Nc1ccc(OCC(=O)NCC2=CC=CN=C2)cc1 is similar to the CID564290 molecule in the PubChem database (Figure 5). Compared to CID564290, the generated new molecule has an additional amide group (-CONH2) [50]. An amide group is a highly electrophilic group that often increases the polarity and water solubility of the compound, thus resulting in lower logP values [51]. It is consistent with our result that the logP decreased from 2.085 to 1.735.

In examples of generated molecules in QED attributes, which are described in Section A.2, the generated CN1C(=O)c2ccccc2NC1CN1CCOC(c2cccnc2)C1 is highly similar to the CID2812011 molecule in the database, as illustrated in Figure 6. They have almost the same substructures and functional groups. However, their groups have different distributions, and the new generated molecule has different carbon chain lengths and side chain directions, which led to a change in the QED [52]. In the example, the QED value increased from 0.895 to 0.927, which achieves the effect of improving drug likeness.

By observing the examples of generated molecules in SAScore attributes in Appendix B, the generated molecule O=C(CCn1cccn1)N1CCN(S(=O)(=O)c2cc(Cl)ccc2Cl)CC1 that is highly similar to the CID72072982 molecule in the PubChem database is shown in Figure 7. However, the SAScore decreased from 2.410 to 2.246, which makes the generated molecule more easy to become a drug. The reason is that the generated molecule reduces the number of stereocenters compared to the CID72072982 one, thus making the main chain (reducing one methyl group) and bond types (olefins) simpler, which reduces the difficulty of molecular synthesis while ensuring high similarity. The phenomenon is also discussed in [38].

In summary, the comparison with the PubChem database shows the similarity between the generated molecules and the recorded molecules. This evaluation indicates that the AC-ModNet has advantages in generating compounds, and the discovered new molecules have potential significance for specific application fields.

## 5. Conclusions and Future Work

This article has proposed a molecular inverse design method, AC-ModNet, which effectively combines the VAE with the GAN. Evaluations in the ZINC dataset prove that the AC-ModNet can efficiently converge, and the generated molecules for the specifying categories are valid, novel, and relatively unique. Comparisons between our model-created molecules and the PubChem database show the applciation value of the AC-ModNet in drug design.

In future work, we will focus on attempting to expand our model in molecular 3D datasets, and we will advance our work in the field of molecular optimization. On the other hand, we will try some new generation models, such as the diffusion model, in the molecular design.

## Figures and Tables

**Figure 1 ijms-25-06940-f001:**
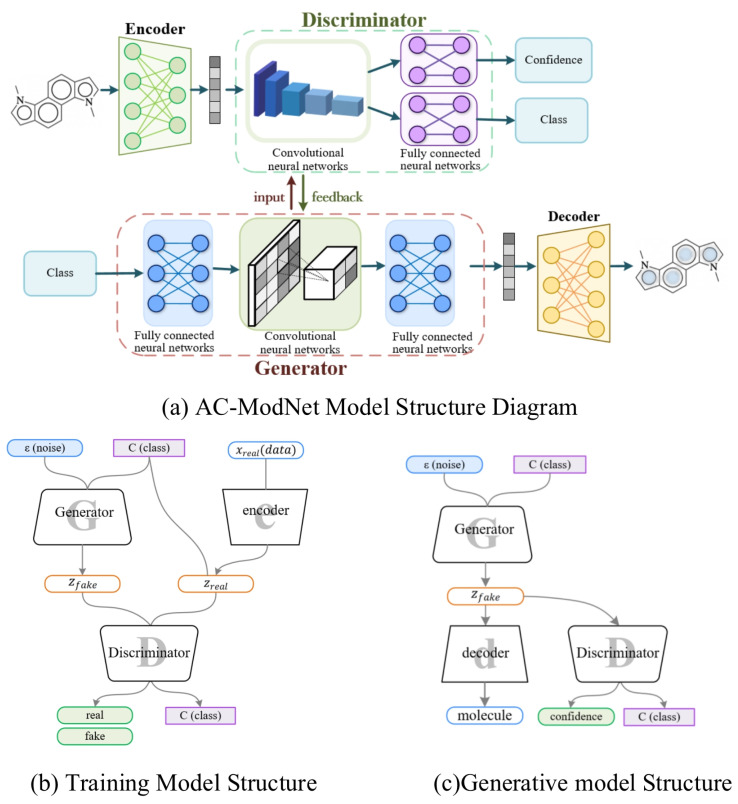
AC-ModNet model structure diagram.

**Figure 2 ijms-25-06940-f002:**
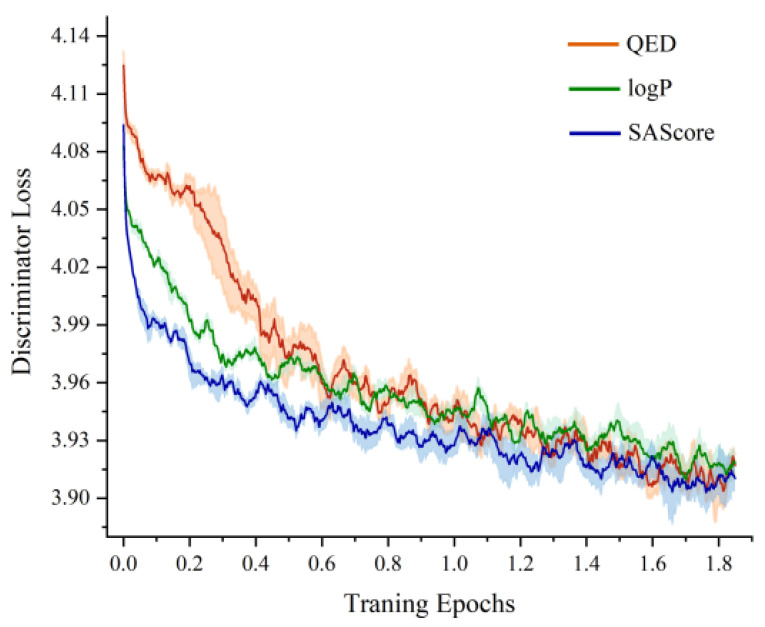
Discriminator loss change curve.

**Figure 3 ijms-25-06940-f003:**
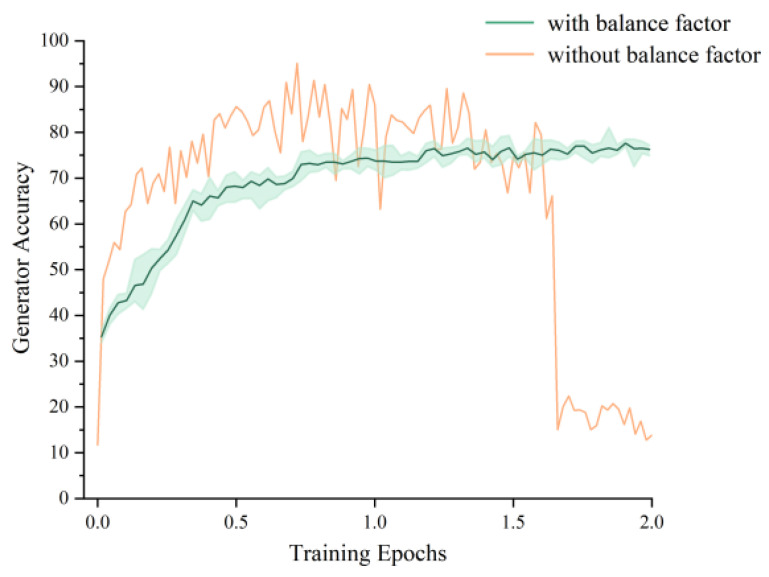
Generator accuracy change curve.

**Figure 4 ijms-25-06940-f004:**
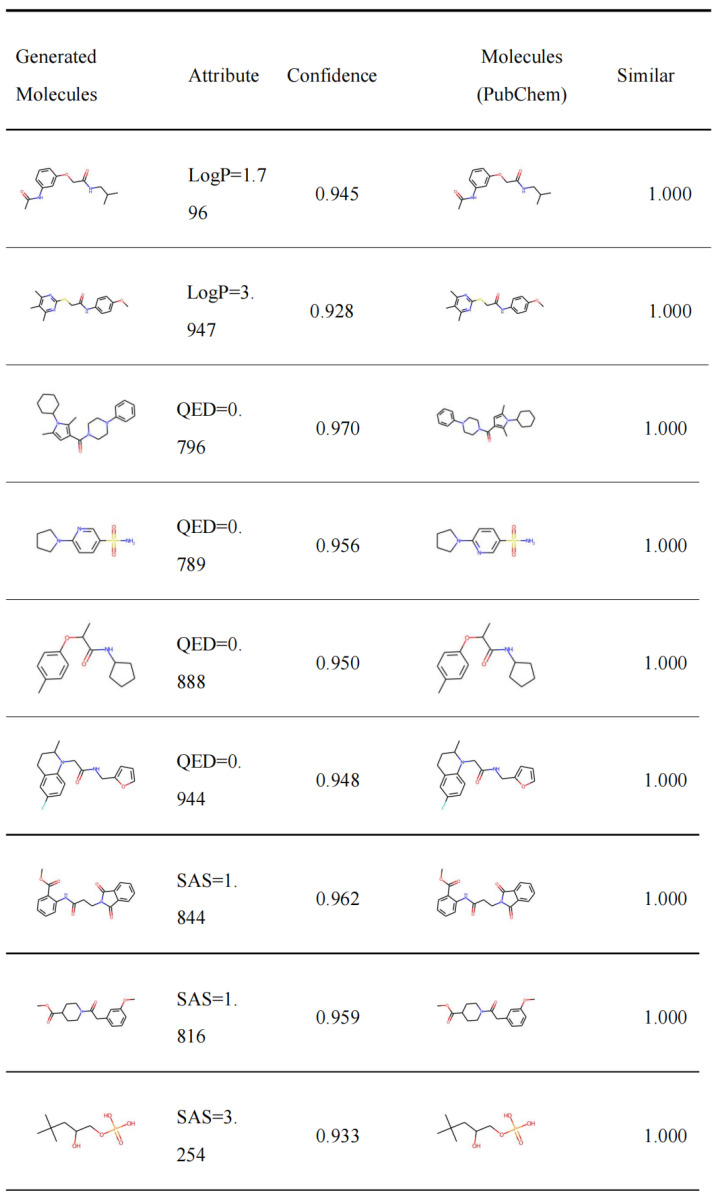
Examples of the molecules generated by our model with real drug molecules in the PubChem dataset.

**Figure 5 ijms-25-06940-f005:**
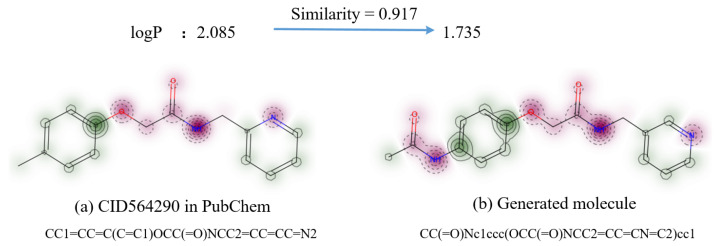
An example of the recorded molecule and the similar generated one in logP.

**Figure 6 ijms-25-06940-f006:**
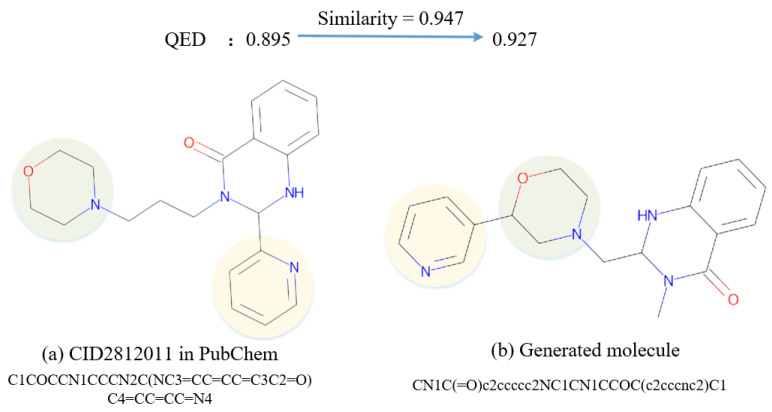
An example of the recorded molecule and the similar generated one in QED.

**Figure 7 ijms-25-06940-f007:**
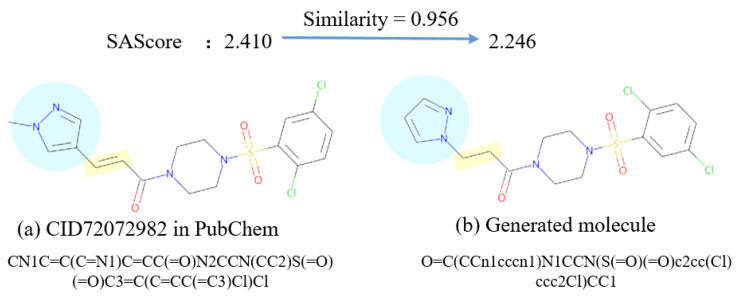
An example of the recorded molecule and the similar generated one in SAScore.

**Table 1 ijms-25-06940-t001:** Attribute data categories and data volumes.

Categories		1	2	3	4	5	6	7
Logp	interval	(−∞,0.0]	(0.0, 1.0]	(1.0, 2.0]	(2.0, 3.0]	(3.0, 4.0]	(4.0,∞)	-
	quantity	13,027	24,026	46,966	67,891	63,386	28,956	-
QED	interval	[0.0, 0.5]	(0.5, 0.6]	(0.6, 0.7]	(0.7, 0.8]	(0.8, 0.9]	(0.9, 1.0]	-
	quantity	20,526	24,026	43,425	69,034	78,351	14,232	-
SAS	interval	(0, 2.0]	(2.0, 2.5]	(2.5, 3.0]	(3.0, 3.5]	(3.5, 4.0]	(4.0, 4.5]	(4.5, 10]
	quantity	13,480	59,794	64,827	46,062	28,967	20,617	16,293

**Table 2 ijms-25-06940-t002:** Comparison of molecular accuracy generated by FCN and CNN.

Categories	Model	1	2	3	4	5	6	7
LogP	model 1	2.94	8.26	11.75	10.73	10.75	8.98	-
	model 2	35.79	31.98	61.79	61.81	37.06	39.2	-
QED	model 1	3.94	8.33	19.0	20.85	18.93	11.12	-
	model 2	35.47	39.11	67.4	71.17	76.61	36.83	-
SAS	model 1	2.69	8.95	19.28	19.52	6.98	10.96	3.05
	model 2	43.08	61.38	64.21	51.45	56.38	44.52	24.74

**Table 3 ijms-25-06940-t003:** Iteration Times of Model Training.

Model	Required Iterations
AC-ModNet	2
Hier-VAE	20

**Table 4 ijms-25-06940-t004:** Property table for generating molecules by specifying attributes.

Categories	Evaluation (%)	1	2	3	4	5	6	7	Avg
LogP	Validity	100	100	100	100	100	100	-	100
	Novelty	98.52	100	97.66	99.78	100	100	-	99.41
	Uniqueness	85.2	61.4	48.6	63.4	59.2	57	-	59.67
QED	Validity	100	100	100	100	100	100	-	100
	Novelty	99.73	97.4	100	99.51	96.98	100	-	98.64
	Uniqueness	62.3	47.9	72.6	57.5	39.7	59.9	-	54.15
SAS	Validity	100	100	100	100	100	100	100	100
	Novelty	98.29	99.56	97.6	99.78	100	99.21	96.76	98.86
	Uniqueness	71.4	63	61.6	89.6	89.4	90.2	67.6	73.59

**Table 5 ijms-25-06940-t005:** Molecular uniqueness and number of generated samples.

Number of Generated Molecules	Uniqueness (%)
1000	77.44
5000	73.17
10,000	65.46

**Table 6 ijms-25-06940-t006:** Model comparison experimental results.

Models	FCD [41,42] ↓	IntDiv1 [43] ↑	IntDiv2 [43] ↑	SNN [44] ↑	Scaf [45] ↑	Frag [46] ↑
VAE [20]	0.099 ± 0.013	0.856 ± 0	**0.85 ± 0**	**0.626 ± 0**	**0.939 ± 0.002**	**0.998 ± 0**
AAE [39]	0.556 ± 0.203	0.856 ± 0	**0.85 ± 0.003**	0.608 ± 0.004	0.902 ± 0.037	0.991 ± 0.0005
JT-VAE [26]	0.3954 ± 0.0234	**0.857 ± 0.0034**	0.8493 ± 0.0035	0.5477 ± 0.0076	0.8964 ± 0.0039	0.9947 ± 0.0002
Hier-VAE [27]	0.439 ± 0.016	0.856 ± 0	**0.85 ± 0**	0.51 ± 0.002	0.92 ± 0.003	**0.998 ± 0.001**
LatentGAN [40]	0.296 ± 0.021	**0.857 ± 0**	0.85 ± 0	0.538 ± 0.001	0.886 ± 0.015	**0.998 ± 0.003**
AC-ModNet (ours)	**0.0881 ± 0.011**	0.855 ± 0	0.84 ± 0.001	0.623 ± 0.001	0.924 ± 0.006	**0.998 ± 0**

## Data Availability

The data presented in this study are available upon request from the corresponding author. The dataset used in this study is sourced from https://www.kaggle.com/datasets/basu369victor/zinc250k (accessed on 16 December 2023).

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
