# Peer review of "AC-ModNet: Molecular Reverse Design Network Based on Attribute Classification"

_ijms, 2024, doi:10.3390/ijms25136940_

Round 1
Reviewer 1 Report
Comments and Suggestions for Authors
This is another generative model architecture proposal, by pure computer scientists - which means that it **MAY** be useful, but the submitted paper fails to prove this. And this is not surprising - you may be outstanding computer scientists, but, sorry to say, you have no clue about chemistry! That is normal - no Homo Sapiens I ever met ever was both an accomplished computer scientist and a good chemist! Therefore, if you wish to work on chemoinformatics, please collaborate with chemoinformaticians and chemists - this is extremely multidisciplinary, and you cannot succeed alone. In the current paper, I cannot fully understand the benefits of your architectures, albeit I work in the field, use and test such models - but I don't develop them myself. What I unfortunately can understand is that you have unwillingly written some pretty nonsensical statements about molecules and chemistry - which make the publication unacceptable in its present form. No, logP does not "follow Lipinski's rule of five" - it is one of the five terms appearing in Lipinski's rules, nothing more. No, improving SAS does not make a molecule "more likely to become a drug".
Which brings us to the core problem of this work: there are many such architectures in use today, and none stands out in terms of actual performance - generating useful, feasible molecules. Now, I do not ask you to actually run a drug discovery project based on your tool and to prove that it found an original compound that can be made by chemists and was active - that is what should be done in principle, but it is not realistic. However, what you certainly need to do is thoroughly benchmark you approach against competitor tools - there is a whole galaxy of benchmarking procedures out there, to the point that review articles are dedicated to the issue on how to benchmark your generative molecule model. See, for example, "De novo molecular drug design benchmarking RSC Med Chem. 2021; 12(8): 1273–1280 ,doi: 10.1039/d1md00074h". Run these benchmarks, then co-opt a chemist on your team, to help you with chemistry questions in the text, and, foremost, discuss on the pertinence of generated compounds from the perspective of a synthetic chemist... because the benchmarks cited are necessary, but far from sufficient: none correctly solves the problem of chemical stability and practical feasibility of generated compounds.
Globally not bad, problems being rather semantic, of meaning: for example, the graph does not contain "molecular structure and chemical information" because "molecular structure" IS nothing but "chemical information" - it's like saying that "Sport and tennis are good for your health"! Proofreading by a native English speaker is definitely a good idea.
Reviewer 2 Report
Comments and Suggestions for Authors
I think this is a fundamentally sound paper that can be published after some clarification.
It is fundamentally an inverse molecular design paper using AC-GAN but used phrases I am not familiar with which slowed down my reading and momentarily confused me at points. For example, the term reverse design is used in title and introduction and inverse design is used in abstract, keywords and conclusion. That was obvious enough to be the same not to cause confusion but felt a bit odd.
The phrases I think need clarifying are:
-- AC which is confirmed deep into the text (p5) as the auxiliary classifier but the term is used before then without definition making it look like it corresponds to the "Attribute Classification" of the title;
-- Attribute Classification itself confused me to start with as I started reading the paper thinking it would all be about classifying attributes and soon realised that attributes were the authors terms for what most other authors call properties or metrics or fingerprints. The authors define what they mean by attributes so it isn't really a problem but having it in the title made it seem more significant than it was. The paper *is* about the model not classifying attributes. Is there any molecular reverse design network not based on attributes?
The paper itself seems a solid application of inverse design to the ZINC dataset. The methodology is sound, and I thought the adaptive learning a strong point, and analysis well defined. However, with all such papers I find myself wondering how one can say this method is superior to others. For example, there is a very similar study "Liu, X., Zhang, W., Tong, X. et al. MolFilterGAN: a progressively augmented generative adversarial network for triaging AI-designed molecules. J Cheminform 15, 42 (2023). https://doi.org/10.1186/s13321-023-00711-1", which really should be cited. Liu et al. use a different model on the same dataset and attributes and make similar claims at the end. It would be interesting to hear the authors comment on how their method compares.
Ultimately, this is a different study and possibly comes up with a different list of interesting molecules and for that may be worth getting an airing.
The paper is quite well-written scientifically but has many moderate English and grammatical errors that would benefit from a native English speaker going through it.
In their definition of "Uniqueness" the authors refer to non-repeatable molecules. I wasn't quite sure what this meant. Do they mean repeated molecules (as in duplicates) or molecules with high similarity or polymer-characteristics (this last seemed highly unlikely but that is the closest in chemical meaning to repeatable).
Round 2
Reviewer 1 Report
Comments and Suggestions for Authors
I am pleased with the answers of the authors to my previous questions, and herewith recommend publication of this manuscript
Comments on the Quality of English LanguageFine
Reviewer 2 Report
Comments and Suggestions for Authors
Thank you to the authors for addressing my comments and clarifying some of my questions.
I still find "2. AC here means attribute classification. In order to solve the task that existing methods cannot generate specific attribute intervals, we proposed a novel AC-ModNet model, which can effectively combine VAE with AC-GAN." a little confusing becase this means AC in AC-ModNet means attribute classification and AC in AC-GAN means Auxiliary Classifier but as the proposed model has both types I guess that's just semantics.
"Uniqueness[9] refers to the proportion of non-repeated molecules" makes sense but the article still refers to non-repeatable molecules.